# Associations Between Common Hip and Knee Osteoarthritis Treatments and All-Cause Mortality

**DOI:** 10.3390/healthcare13172229

**Published:** 2025-09-05

**Authors:** John W. Orchard, L. Edward Tutt, Anna Hines, Jessica J. Orchard

**Affiliations:** School of Public Health, University of Sydney, Physics Rd, Sydney, NSW 2006, Australia; ltut2703@uni.sydney.edu.au (L.E.T.); annahines00@gmail.com (A.H.); jessica.orchard@sydney.edu.au (J.J.O.)

**Keywords:** osteoarthritis, arthroplasty, exercise, GLP-1 RAs (glucagon-like peptide-1 receptor agonists), NSAIDs (non-steroidal anti-inflammatory drugs), opioids, mortality

## Abstract

**Background:** Osteoarthritis has a large and growing burden in an ageing population. Controversy exists in current management, particularly regarding opioid use due to increasing negative effects. Clinicians need guidance on the individual mortality associations for common osteoarthritis treatments when compared to a control. **Aims:** The aim is to undertake a structured narrative literature review comparing mortality associations for common osteoarthritis management options. **Methods:** A search strategy (Web of Science 23 September 2024) was performed to identify observational studies which reported all-cause mortality in a treatment group compared to a control. The control group could be either the general population or those with osteoarthritis who were treated with the following: NSAIDs (non-steroidal anti-inflammatory drugs), opioids, paracetamol, GLP-1 RAs (Glucagon-like peptide-1 receptor agonists), hip or knee arthroplasty, or exercise. Articles were screened by two authors, and each included article was assessed for adequate quality using the strengthening the reporting of observational studies in epidemiology (STROBE) framework. **Results:** Of 2362 studies retrieved, 39 cohort studies met the inclusion requirements. Exercise, compared to no or lower levels of exercise, had ten studies reporting substantially reduced all-cause mortality. GLP-1 RA agonists had two related studies showing all-cause mortality reduction up to 5 years. Mortality following joint arthroplasty followed a multi-phasic response. There was a short-term post-surgical increase in mortality. However, from 90 days post-surgery to 8–11 years, there were significant reductions in mortality. After 9–12 years post arthroplasty, mortality increased and became significantly higher. Opioids were associated with an increase in mortality in 6 out of 7 studies. Inconsistent trends were found for NSAIDs and paracetamol. **Conclusions:** Exercise and GLP-1 RA prescription are associated with reduced all-cause mortality. Arthroplasty was found to have survival benefit until 9–11 years post-operatively, whereafter mortality then increased. Opioids were found to consistently increase mortality when used for non-cancer pain at all time points. The other common osteoarthritis treatments assessed were not consistently associated with changes in mortality.

## 1. Introduction

Osteoarthritis (OA) is one of the most prevalent chronic diseases, affecting approximately 8% of the world’s population [1]. This percentage is reflected in Australia, with almost 2.1 million Australians having OA according to Australian Institute of Health and Welfare estimates in 2022 [2]. OA also has a large healthcare cost and burden of disease, estimated to account for 2.5% of the total disease burden in Australia. Between the years of 2020 and 2021, OA accounted for an estimated AUD 4.3 billion spent [2]. The prevalence of OA is increasing, associated with an ageing population. There was a 38% increase in hip and knee osteoarthritis presentations managed by Australian General Practitioners in the years 2010–2016 compared to 2005–2010 [3].

There is no ‘cure’, and therefore management of this chronic condition aims to relieve pain, improve function, and restore quality of life [4]. Generally, guidelines recommend a stepwise approach to management, initially with education, exercise, and weight management to precede topical or oral analgesia. Surgical intervention (arthroplasty) is recommended in severe cases that are unresponsive to conservative treatment [5]. The use of paracetamol and particularly opioid medications is controversial, with conflicting recommendations across guidelines. Non-steroidal anti inflammatory drugs (NSAIDs) are generally recommended by most guidelines for OA, but are known to cause side effects and hence contra-indicated for many patients with co-morbidities [6]. Hunter and Bowden [7] lament that clinicians mismanage OA, under-utilising efficacious and evidence-based lifestyle behaviour management strategies and weight loss in favour of other less effective, more costly, and potentially harmful treatments.

Historically, opioids were recommended for refractory pain in hip and knee OA [8]. While American guidelines still currently recommend tramadol (a ‘weak’ opioid), Australian and UK guidelines do not [9,10], in response to evidence suggesting that treatment regimes involving opioids are neither effective nor cost effective, and with potentially severe side effects [11]. Regardless, opioids remain widely prescribed, with an estimated 1 in 5 to 1 in 10 Australian patients with OA receiving at least one opioid prescription as part of their management [12]. In the US, the proportion of elderly OA patients being prescribed opioids at least once increased from 31% in 2003 to 40% in 2009 [4]. It is estimated that by the year 2030, there may be a three-fold increase in opioid use among OA patients [13].

Similarly, the role of paracetamol (acetaminophen) in management of OA is unclear. Many guidelines recommend paracetamol as a first-line pharmacological agent; however, there are exceptions [8]. For example, Australian guidelines only recommend paracetamol ‘conditionally’, for use as a short-term trial, due to emerging doubt regarding paracetamols effectiveness, particularly compared to NSAIDs, for the symptomatic management of OA [10]. Its harms, traditionally considered minimal, can include cardiovascular adverse events, gastrointestinal adverse events, and renal impairment [14].

An emerging off-label treatment for knee OA in overweight patients is Glucagon-like peptide-1 receptor agonists (GLP-1 RAs), as not only have these been proven to be effective for weight loss (which indirectly should help lower limb OA), there is already one high-quality randomised trial showing actual reductions in knee pain [15].

Surgery (in the form of joint arthroplasty) is uniformly recommended across the guidelines for those with severe symptoms affecting quality of life, for which non-surgical management is unsuitable or ineffective [5]. However, cost effectiveness of these surgeries has been questioned with contention over their mortality benefit in the short and long term [16].

In light of these controversies, it is important to review the all-cause mortality for these common OA treatments. We chose a structured narrative review format which allowed for a wider inclusion and interpretation of variable study designs, patient samples, interventions, measures, methodological shortcomings, and paucity of literature.

### Aims

Our review aimed to clarify the safest choices for clinicians in treating this common and debilitating disease amongst some of the most common treatments. The primary aim was to undertake a structured narrative literature review reporting the mortality associations of non-steroidal anti-inflammatory drugs (NSAIDs), Glucagon-like peptide-1 receptor agonists (GLP-1 RAs), exercise, arthroplasty, paracetamol, duloxetine (a SNRI (serotonin-norepinephrine reuptake inhibitor)), and opioids, each against a control population, for comparison in the context of osteoarthritis management.

## 2. Methods

### 2.1. Literature Search

A search was conducted of the Web of Science database on 23 September 2024 with strategy detailed in Table 1. We chose Web of Science as an umbrella database that contains multiple other databases within it, including Medline/PubMed.

Observational studies were chosen as they are an appropriate methodology to assess associations of treatment on mortality and major morbidity. For each treatment, if multiple cohort studies were included, then any case–control or cross-sectional study was not considered, and if multiple all-cause mortality associations were reported from the same study population, only the first report was included.

#### 2.1.1. Inclusion and Exclusion Criteria

For inclusion, studies needed to be of an observational study design, evaluating one of our chosen osteoarthritis treatments against a control, with an outcome of all-cause mortality. If there was mention for the treatment indication, osteoarthritis, obesity, or chronic non-cancer pain were included. Control populations included patients with the same conditions that did not use the treatment, or the general population if there was not a matched population available.

Studies were excluded if treatment was used for a different indication than above, for instance depression, where doses and duration can be different [17]. Similarly, articles were excluded if the study cohort was limited to a specific population with pre-existing conditions, such as chronic kidney disease or coronary artery disease, as currently, guidelines carry cautionary use or state contraindications for certain treatments [17]. For instance, the Australian Government’s Therapeutic Goods Administration (TGA) advises health professionals to avoid prescription NSAIDs in patients with “previous myocardial infarction, angina, cardiac failure, hypovolemia, significant peripheral vascular disease and pre-existing significant renal/liver dysfunction”.

We also excluded articles for which the full text was unavailable or studies only published as conference abstracts.

#### 2.1.2. Data Extraction and Quality Assessment

Title and abstracts were independently screened by two authors (LET and AH). After this, an eligibility screening was conducted by one author (LET or AH). Each included paper was assessed following the Strengthening the Reporting of Observational Studies in Epidemiology (STROBE) checklist [18], consisting of a total of 22 checklist items with important consideration of case and control selection and criteria, matching, comparability between groups, and adjustment for potential confounders. The STROBE guidelines were created to ensure high quality presentation of observational studies, and is appropriate given our search is focusing on observational studies [19].

The following information was extracted from included studies: treatment, study name, study site, study design, cases, duration of follow up, measure of effect, outcomes, exposure, and results. Screening and selection of the studies were performed using RAYYAN, an online systematic review management tool (https://www.rayyan.ai/ (URL accessed on 23 September 2024)) which allows for collaborative screening of studies. Included studies were required to report on all-cause mortality, comparing the intervention or exposure of interest to a control group or the general population. As such, studies comparing one OA treatment directly to another treatment were excluded, as were studies focusing on a subset other than the OA population, for example, patients with chronic kidney disease. To examine studies on analgesia types (opioids, paracetamol, duloxetine), we included studies that focused on non-cancer chronic pain (of which OA is a common cause) rather than OA specifically, due to paucity of specific studies.

## 3. Results

The search identified 2362 records, and 5 duplicates were removed. Figure 1 shows the PRISMA flow chart. There were 82 studies that warranted full text retrieval with exclusion reasons provided on the 43 studies that were then excluded after full text assessment. A final 39 studies were included, all of which were cohort studies.

### 3.1. Quality

Due to their observational nature, all studies were deemed low quality with moderate-high risk of bias; however, no serious omissions were noted. All studies reported either hazard ratios (HRs), Relative Risk (RR), or Standardised Mortality Ratios (SMRs), with the index treatment being compared to a control of no treatment or the general (age-matched) population. This allowed consistency in reporting results which we split by treatment. Whilst there are subtle differences between HR, RR, and SMR and the various studies all tended to have slightly different definitions, we present the results as ratios assuming they have a consistent meaning of the risk/rate of the treatment population dying over the observation period compared to the risk/rate of the control population dying. We also consistently report 95% confidence intervals (CIs) with the ratios and call a ratio a significant finding if the 95% CIs do not include 1.0.

### 3.2. Exercise

Ten studies reported an intervention of exercise, with duration of follow up ranging between 2 years and 16 years, with a cohort size between 364 and 167,413 (Table 2).

Eight of the studies reported exploration into the frequency of exercise and the relationship with all-cause mortality. Six of these studies reported statistically significant findings that when compared to those with the lowest or no reported levels of exercise; the higher frequency of exercise provided an all-cause mortality benefit [22,23,25,28,29,30]. This trend, however, was not universal in the studies, for instance in the study by Ekblom-Bak et al., which reported data on self-reported non-exercise physical activity [21]. When compared to subjects who reported the lowest third of exercise frequency, the middle tertile of subjects had unclear results for mortality benefit [21]. There was, however, a statistically significant finding if the comparison was with the highest tertile of respondents, a reduced all-cause mortality of HR of 0.7 (0.53–0.98) [21]. Interestingly, a different result was noted in the study by de Boer et al., which compared a control population that did not participate in moderate-to-vigorous physical activity (MVPA) to subjects were divided into quintiles based on MVPA minutes/week [20]. A statistically significant all-cause mortality benefit was noted for increasing frequency of MVPA; however, this benefit was lost in the highest quintile [20].

The finding that the cohort with the highest levels of exercise may not have an all-cause mortality benefit is similarly seen in the study by Moshkovits et al., which compared a level of intensity of exercise [27]. This study reported an all-cause mortality benefit for light-moderate exercise; however, it also reported an uncertain effect on vigorous exercise, when compared to subjects reporting no or past exercise [27].

Besides frequency and intensity, consistency of exercise is a factor considered in the study by León-Muñoz et al. [24]. In this study, a lack of exercise was defined by sitting time (ST) and followed over a 2-year period. When compared to their peers in the study, the subjects who were below the median of ST at the two time points, had a reduced all-cause mortality with a HR of 0.75 (0.62–0.90) when compared to those who had a higher ST at both time points [24].

### 3.3. Glucagon-like Peptide-1 Receptor Agonists

The intervention of Glucagon-like peptide-1 (GLP-1) receptor agonists had two studies meeting inclusion criteria (Table 3). Both studies reported in 2024 derived data from the TriNetX Global network; however, they reported different inclusion criteria, duration, and primary outcome. Alenezi et al. [31], had a duration of only 6 months, with a study population of 15,655 cases, while Huang et al., had a 5-year duration of follow-up and consisted of 12,123 cases. Both showed reduced hazard ratios for GLP-1 RA prescription compared to no use, Huang et al. [32], reported HR 0.23 (0.15–0.34), while Alenezi et al. [31] reported HR 0.27 (0.21–0.34).

### 3.4. Non-Steroidal Anti-Inflammatory Drugs

The all-cause mortality outcome of oral non-steroidal anti-Inflammatory drugs (NSAIDs) compared to a control reported inconsistent results (Table 4). Four studies were included with duration between 30 days and 10 years, and had between 1025 and 649,339 cases. Three of the studies reported no statistically significant difference in mortality outcomes, while only one study, Han et al., reported in 2020 a statistically significant result of reduced all-cause mortality HR 0.84 (0.83–0.85). There were no included studies that reported all-cause mortality data on topical NSAIDs. The results were considered inconsistent as the largest cohort with the longest follow-up showed potentially a slight protective effect of NSAIDs, whereas a population-specific study (knee osteoarthritis) showed a strong trend (not quite reaching statistical significance) of increased all-cause mortality.

### 3.5. Arthroplasty

A total of 14 papers were included that reported all-cause mortality associated with knee or hip arthroplasty (nine examining knee arthroplasty alone, four examining hip arthroplasty alone, and one examining both). There were no relevant studies on other surgical treatments for OA. Mortality was not sufficiently assessed by our methodology in the immediate post-operative period (although it is quite possible/likely that mortality may be increased in the immediate fortnight post-surgery). Mortality benefits (decreases) after arthroplasty were well-established by 90 days but diminished and even reversed many years later.

#### 3.5.1. All-Cause Mortality Following Hip Arthroplasty

A total of 368,086 cases were included, with follow up ranging from 90 days to 21 years (Table 5). Most compared those with OA undergoing hip arthroplasty with cases matched from the general population. 

Maradit Kremers et al. [37] and Lie et al. [38] gave similar results for SMR following hip arthroplasty for OA, 0.82 (0.76–0.88) and 0.81 (0.79–0.83), respectively, suggesting there is a mortality benefit for this OA treatment. In the Lie study, age-adjusted mortality was increased compared to the general population for patients younger than 60 years. Cook et al. [39]’s cohort study focused on shorter term outcomes, with mortality benefit shown both at 60 and 90 days post-operatively. Cnudde et al. [40] and Gordon et al. [41] show similar survival benefit longer term, but only up until 12 years and 8.8 years, respectively, where mortality increased.

**Table 5 healthcare-13-02229-t005:** All-cause mortality associations following total hip arthroplasty; significant findings at 95% CI ratio in bold.

Authors	Cases	Follow Up Mean (Range)	Control	Results (Ratios and 95% CIs)
Cnudde et al., (2018) [40]	131,808	5.9 years (1–14)	General population	**At 1 year: R = 1.01 (1.01–1.0)** **At 5 years: R = 1.03 (1.03–1.03)** **At 10 years: R = 1.02 (1.02–1.03)** At 12 years: R = 1.01 (0.99–1.02)
Cook et al., (2022) [39]	103,563	90 days	OA population	30 days: 1.05 (0.91, 1.23) **60 days: 0.82 (0.73, 0.92)** **90 days: 0.68 (0.62, 0.76)**
Gordon et al., (2016) [41]	91,527	10 years (7–21)	General population	Overall 1.00 (0.99–1.01) **5–9 years 0.90 (0.88–0.92)** **9–13 years 1.05 (1.03–1.08)** **13–17 years 1.11 (1.07–1.14)** **17–21 years 1.19 (1.13–1.26)** Crossover occurred at 8.8 years (CI 8.3–9.3)
Kremers et al., (2016) [37]	1645	11.9 years	General population	**0.82 (0.76–0.88)**
Lie et al., (2000) [38]	39,543	5.2 years (0–10.4)	General population	**All THR patients 0.81 (0.79–0.83)** **OA patients 0.68 (0.66–0.70)** **First 60 days 1.39 (1.25–1.56)**

#### 3.5.2. All-Cause Mortality Following Knee Arthroplasty

A total of 480,057 cases were included (Table 6). Most studies were large, with the smallest cohort being Kim et al. [42] with 601. Total follow up ranged from 90 days to 33 years.

Overall, a mortality benefit was seen in the majority of studies with SMR ranging from 0.61 in Choi et al. [43] to 0.868 in Maradit Kremers et al. [37], with benefit seen as early as 60 days post-operatively in Cook et al. [39]. At later years post arthroplasty however, again a mortality increase is seen after 10 years in Visuri et al. [46] and after 11 years in Zhou et al. [48].

#### 3.5.3. Opioids

A total of seven studies were included with a total of 13,085,732 cases. Follow up ranged from 0 to 11 years (Table 7).

No studies focused on opioid use in OA patients specifically, rather, all studies examined a broader population of those with non-cancer pain. Consistently, opioid use for chronic non-cancer pain resulted in increased mortality. This effect was increased when strong opioids were used, or opioids were used chronically. Only Oh et al. [53] showed any mortality benefit, for chronic weak opioid use, also finding a harm for stronger opioid usage in the same study

#### 3.5.4. Paracetamol

Only two studies were included, comprising 55,314 cases with follow up spanning 0–8 years (Table 8). Girard et al. [56] showed no mortality difference in those that took paracetamol compared with matched controls. Conversely, Lipworth et al. [57] found a significantly increased mortality compared with the general population, particularly in the first year of paracetamol prescription SMR 2.9 (2.8–3.0) although this study may have had significant confounding effects.

#### 3.5.5. Duloxetine

No relevant studies were identified. Although we did include duloxetine in the search strategy, we have not discussed this treatment option due to our search not revealing papers with data on relative mortality.

## 4. Discussion

This study found two osteoarthritis treatments with benefits related to all-cause mortality (exercise and GLP-1 RAs), one treatment with harm related to all-cause mortality (opioids), one treatment with time-dependent effects (arthroplasty with short-to-medium-term benefits that eventually reverse in the longer term) and other treatments with no consistent associations, such as paracetamol and NSAIDs. An important consideration for prescribers is that this review does not assess effectiveness of treatments, which clearly is also of importance in clinical decisions.

### 4.1. Exercise

Exercise had both a high number of studies and consistency of findings showing the major benefit of exercise versus no exercise in terms of all-cause mortality, particularly with respect to moderate exercise. A limitation is that the exercise studies were not specific to treatment of osteoarthritis. There was a reported finding in two studies, de Boer et al., and Moshkovits et al., that suggested a possible reverse “J” shaped effect of mortality benefit as frequency and intensity of exercise increases, which has been widely reported [58,59,60]. Multiple proposed mechanism have been offered to explain these findings, especially in the context of endurance athletes, such as increased rates of arrythmias and ventricular remodelling [61,62]. Generally high levels of exercise are thought to be safe; however, in very high levels, it is perhaps not as beneficial as moderate levels of exercise.

It is also worth noting that elite endurance athletes, despite the potential of above concerns, have an increased life expectancy when compared with the general population [63]. In terms of treatment of knee osteoarthritis, it is also highly unlikely that anyone would be prescribed high enough level exercise for this condition (e.g., ultramarathon running) that they would enter the second upward slope of any U-shaped curve.

The overall benefit of exercise is likely due to the known impact exercise has in reducing cancer and chronic disease, including improvements in metabolic and cardiovascular disease [64,65]. There exist multiple possible mechanisms for these findings and can be differentiated through exploring the effects of exercise, by both increasing physical activity but also reducing sedentary behaviour. Increasing physical activity has been proposed to protect against adverse metabolic consequences of stressors and chronic disease through positive biological and psychological pathways, having been shown to increase immune function and reduce excessive inflammation [66,67,68]. Conversely, sedentary behaviour has been linked to the skeletal muscle loss (sarcopenia) in older adults, through pathways such as mitochondrial dysfunction and oxidative stress, by leading to chronic systemic inflammation and cellular deterioration and death [69].

### 4.2. GLP-1 RAs

The significant all-cause mortality benefit resultant from GLP-1 RA prescription reported in both included studies has multiple components to consider. The GLP-1 RA drug class is an increasingly studied therapeutic option, having significant metabolic effects, primarily for improving glycaemic control, but now also over multiple organ systems [70]. Of particular relevance to this narrative literature review, is robust evidence this drug class can contribute to significant reduction in cardiovascular events. For instance the double-blind randomised control trial, Liraglutide and Cardiovascular Outcomes in Type 2 Diabetes (LEADER) study, showed liraglutide reduced cardiovascular mortality compared to a placebo for those with type 2 diabetes mellitus (T2DM) and at high-risk of cardiovascular disease (CVD) HR 0.87 (0.78–0.97) [71]. The two included studies in this review are part of the exploration into increasing the knowledge of safety profile, in particular to patient without T2DM, which will have greater impact in the general osteoarthritis management [32]. Further trials are warranted to examine extent of mortality reductions in the OA specific population with GLP-1 agonists.

### 4.3. NSAIDs

The lack of consistent findings for NSAIDs is unsurprising, as there remains controversy for their risk. NSAIDs work by inhibiting two recognised cyclooxygenase (COX) pathways, COX-1 and COX-2, with drugs with varying affinities to both [72]. COX-2-selective NSAIDs have been associated with increased vascular events; however, the non-selective NSAIDs offer greater uncertainty, with the suggestion that different drugs in this class offer their own individual risk profiles [73]. The one study with reported all-cause mortality benefit [34] was likely to be due to differing prescribing patterns, a form of allocation bias, where fewer prescriptions were given to patients with worse renal or cardiovascular disease.

We have concluded that there is no consistent relationship between NSAIDs and all-cause mortality in the studies we reviewed according to our inclusion criteria. Although not considered by our study, it seems likely that the effects of NSAIDs are different across different systems. Multiple large nationwide studies, from Denmark and the United Kingdom have found an association between NSAIDs and increased cardiovascular mortality [74,75,76]. However, some other studies have shown a protective effect of NSAIDs against development of or progression of certain cancers [77,78,79]. These two discordant associations (from the two leading causes of death) might be responsible overall for non-significant all-cause mortality associations.

Additionally, this narrative literature review did not explore the effect of dosage on mortality, and often relied on prescription data, as opposed to medication actually taken, which all can have important impact on reducing the quality of the findings.

### 4.4. Arthroplasty

Joint arthroplasty was shown to both increase and decrease mortality depending on the time since operation. Although not a finding of our review, it is well established that immediately post-operatively, mortality rates increase for a short time due to peri-operative complications and increase in surgical stress [43]. In our studies, from as early as 60 days, through to 8–11 years post-operatively, a mortality benefit was seen, suggesting that the (vast majority of) patients who survive the surgical encounter go on to live longer over the next decade. Multiple reasons have been suggested for this intermediate mortality benefit, from decreasing or eliminating need for harmful NSAIDs and opioids [80], to increased ability to exercise and participate in activities of daily living [43]. However, it may also represent selection bias favouring ‘healthier’ patients for surgery and anaesthesia [81], or it is possible that interventions in preparation for surgery, for example, pre-operative assessment, prehabilitation, and increased monitoring may explain a beneficial impact on reducing mortality [39] in those who had arthroplasty. All studies in our review attempted to address this by adjusting for multiple comorbidities; however, due to their observational nature, only known confounders can be controlled for [82], meaning confounding likely persists.

Interestingly, studies with long term follow up found an increase in mortality that occurred at 9–12 years post-operatively. This may reflect mortality associated with a decrease in function at the end of the arthroplasty’s lifespan, mortality associated with revision arthroplasty, an increased mortality rate observed in those who require arthroplasty at a younger age, or mortality associated with the multi-inflammatory state of OA [40,45,46].

### 4.5. Opioids

Our review confirmed that opioids are associated with increased mortality. Unsurprisingly, stronger opioids used over a longer time period increased this mortality risk further. Indication bias is likely present, where the reason for needing strong analgesia is mortality association, rather than the analgesia itself [83]. One study found equivocal mortality benefit with chronic weak opioids (codeine or tramadol), which may not be associated with serious adverse effects of opioid use such as cancer, endocrinopathies, heart disease and immunosuppression [53]. This also may represent publication bias; the bias towards the selective publication of studies reporting adverse effects related to opioids which is increasingly seen as a consequence of the opioid epidemic in America [82]. While the type of opioid and duration of prescription were known in the included studies, dose and patient compliance were not—that is, we cannot say how much opioid patients were actually taking. This significantly affects the validity of results.

### 4.6. Paracetamol

While paracetamol is widely prescribed and used, the available literature supports perhaps a very small mortality risk due to paracetamol, likely due to its mechanism as a cyclooxygenase inhibitor [14,80]. In our review, only two studies reported on mortality associated with paracetamol, each reporting differing results. One cohort studied which found no mortality association were vigorously matched [56], with extensive modelling undertaken to reduce the influence of confounders. Lipworth et al. [57] note the likely effect of indication bias in their study, where mortality was significantly increased likely due to serious or chronic illness. We therefore found no consistent association between paracetamol and mortality but cannot rule out a small increase.

### 4.7. Strengths of This Review

The strength of our review lies in the inclusion of ‘real world’ prospective cohort studies with large sample size, increasing applicability of the findings. Observational trials often provide the best source of information for situations where large sample sizes require long-term follow up to capture mortality outcomes [84]. Our structured narrative review enabled the inclusion of a broad range of studies that helped compare relative mortality rates between different treatment options. This is the first review that has directly compared the mortality ratios of the most common treatments for osteoarthritis with each other. Notwithstanding multiple limitations, which we detail following, the results for most of the major treatment types have generally shown consistent trends, suggesting the validity of the review.

### 4.8. Limitations of This Review

Studies based on ‘real-world’ clinical information are known to be at risk of multiple biases, including differential recording, misclassification, and missing data [85]. With no RCTs included, exposures in all studies were not randomly assigned, therefore relying on propensity scores to account for potentially confounding factors [56], and risking still having many residual confounders. The variation between studies where some used a non-treatment osteoarthritis population as a control group and others used the general population adds to possible confounders. It is likely that the mortality influence of NSAIDs, opioids, paracetamol and other over-the-counter medications were all present in studies examining arthroplasty, for example, which might impact the validity of results. The significant influence of indication bias throughout can also not be understated. We therefore acknowledge the limitation of observational studies in the assessment of moderate effects due to inherent potential for moderate or large bias [86]. A key benefit of observational studies, particularly when exploring mortality as an outcome, is there generally longer follow up, less restrictive study population and larger sample sizes, offering potentially improved external validity compared to RCTs and meta-analyses [87]. Being based on observational studies, the review provides a low-moderate quality of evidence, and a low number of included studies for some treatments, especially regarding the newest option, the GLP-1 RA drug class. A further limitation is that we only reviewed papers that were part of Web of Science. Whilst Web of Science is an umbrella database that includes papers from multiple other databases, there may have been relevant studies in other databases or grey literature that may have been relevant and that we have missed. Our search strategy terms may also have missed other relevant studies. Our narrative review methodology was not a robust as a systematic review, although a systematic review would have been challenging because of the diverse types of treatments.

With respect to the pharmacotherapy options, other than for the analysis of opioids (where some subgroup analysis has been presented which differentiates between strong and weak opioids), there was no data available in our review which took dose and frequency into account. It is very plausible that certain medications might be “safe” (that is having no negative mortality association) in low and infrequent doses, but less safe (being associated with increased mortality in higher regular doses), and our review was unable to analyse this.

Duloxetine was unable to be evaluated in this review due to paucity of studies. While conditionally recommended in guidelines [10,88], it lacks approval for this indication in most jurisdictions. Studies may become available in the future if its use increases.

Our review did not assess efficacy of treatments; however, this is a crucial additional consideration for prescribing clinicians. Another important consideration, rather than the blunt tool of all-cause mortality, is a nuanced consideration of quality of life and years spent free from disability.

## 5. Conclusions

Exercise and GLP-1 RAs were OA treatments associated with mortality benefits. Arthroplasties (hip and knee) were also found to have patient survival benefits until 8–11 years post-operatively, whereby mortality then increased compared to the general population. Opioids were generally found to increase mortality when used for non-cancer pain. Findings were inconsistent regarding NSAIDs and paracetamol due to limitations of the available literature.

Our findings suggest clinicians could consider reducing opioid prescribing for patients with OA simply on the basis of higher mortality rates, in favour of less harmful treatments such as lifestyle change, exercise, and weight management. Arthroplasty should still be considered an important treatment for end stage OA, particularly for elderly patients with expectation of moderate life expectancy (for example, ages 65–85). Caution should be applied in recommending arthroplasty for younger (middle-aged) patients who are more likely to be subject to the later increase in mortality. A limitation of our recommendations is that they are based only on our observations of association with mortality rates, without taking into account relative treatment efficacy. Further studies are warranted to examine the relative safety and association with mortality of non-opioid painkillers such as NSAIDs and paracetamol.

## Figures and Tables

**Figure 1 healthcare-13-02229-f001:**
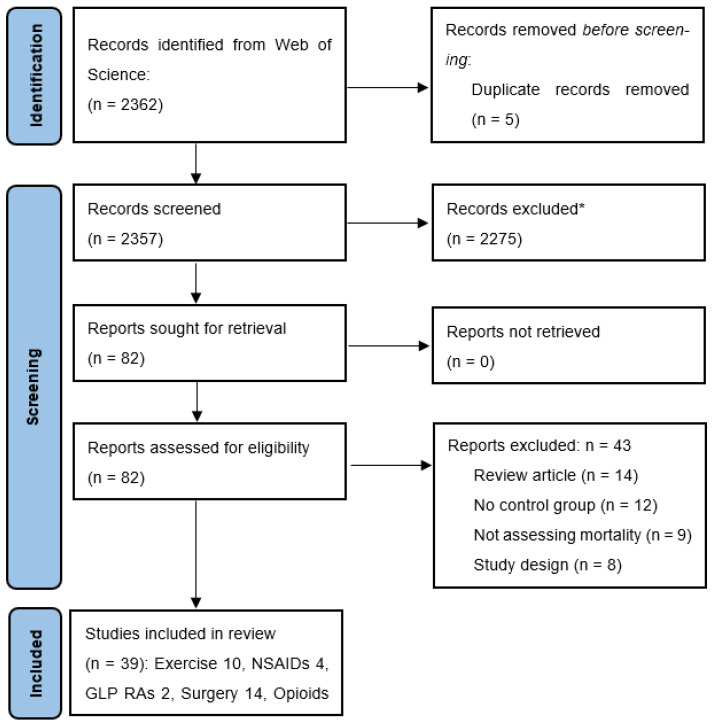
Flow diagram for study review and selection. * Exclusion criteria included animal studies, human studies with different exposures or not reporting on mortality.

**Table 1 healthcare-13-02229-t001:** **Search Strategy (Web of Science).**

#1	(All cause mortality OR cardiovascular mortality OR life expectancy OR mortality)
#2	(Cohort OR cross-sectional OR case–control)
#3	#1 AND #2
#4	(GLP-1 agonists OR Glucagon-like peptide-1 receptor agonists)
#5	(Exercise OR Physiotherapy OR Exercise Physiology)
#6	(Opioid OR Opiate)
#7	(Arthroplasty OR Hip Replacement OR Knee Replacement)
#8	#4 OR #5 OR #6 OR #7
#9	(Obesity OR Osteoarthritis OR Chronic non-cancer pain)
#10	#8 AND #9
#11	(NSAID OR NSAIDs OR Paracetamol OR Acetaminophen OR Duloxetine OR SNRI)
#12	#10 OR #11
#13	#3 AND #12

**Table 2 healthcare-13-02229-t002:** Associations of exercise with all-cause mortality; significant findings at 95% CI ratio in bold.

Authors	Cases	Duration of Follow-Up (Average)	Population and Exposure	Results (Ratios and 95% CIs)
de Boer et al. (2023) [20]	84,230	7.8 years	Age between 25 and 75 years, without T2DM, CVD or cancer at baseline. Moderate-to-vigorous physical activity (MVPA) minutes/week score quintiles	No MVPA: 1.00 (reference) Practicing MVPA: **0.77 (0.64–0.93)** By MVPA quintiles: Q1: 0.82 (0.65–1.04) **Q2: 0.77 (0.60–0.99)** **Q3: 0.71 (0.55–0.91)** **Q4: 0.73 (0.57–0.92)** Q5: 0.82 (0.66–1.02)
Ekblom-Bak et al. (2014) [21]	4232	12.5 years	Intake 60 year olds. Self-reported frequency on questionnaire of non-exercise physical activity (NEPA) score tertiles	Low NEPA: 1.00 (reference) Moderate NEPA: 0.85 (0.67–1.08) **High NEPA: 0.7 (0.53–0.98)**
Inoue et al. (2008) [22]	83,034 (Men *n* = 39,183; Women *n* = 43,851)	5 years	45–74, daily total physical activity (PA) level by metabolic equivalent of task (METs)/day score quartiles	Men Lowest: 1.00 (reference) **Second: 0.76 (0.68–0.83)** **Third: 0.79 (0.71–0.87)** **Highest: 0.67 (0.61–0.74)** Women Lowest: 1.00 (reference) Second: 0.70 (0.62–1.18) **Third: 0.61 (0.53–0.70)** **Highest: 0.54 (0.52–0.82)**
Landi et al. (2008) [23]	364	2 years	Age > 80. Sedentary defined as subjects walking less than 1 h/day, active walking ≥1 h/day	Sedentary: 1.00 (reference) **Active: 0.36 (0.12–0.98)**
Leon-Munoz et al. (2013) [24]	2635	2 years	Age > 60. Sedentary behaviours defined by comparing subjects self-reported sitting time (ST) behaviour at 2 points in time (2001 and 2003). Consistently sedentary (>Median ST in 2001 and 2003) Newly sedentary (≤Median ST in 2001 and >Median in 2003) Formerly sedentary (>Median in 2001 and ≤Median in 2003) Consistently non-sedentary (≤Median in 2001 and 2003)	Consistently sedentary: 1.00 (reference) Newly sedentary: 0.91 (0.76–1.10) Formerly sedentary: 0.86 (0.70–1.05) **Consistently non-sedentary: 0.75 (0.62–0.90)**
Min et al. (2020) [25]	167,413	5.4 years	Age 40–79. PA ‘sufficiency’ according to the International Physical Activity Questionnaire (IPAQ) [26]	Insufficient PA: 1.00 (reference) **Sufficient PA: 0.85 (0.82–0.88)**
Moshkovits et al. (2024) [27]	1210	13 years	Age 65+. Baseline self-reported physical activity intensity. Light-moderate (household activities, walking, yoga, dancing, Feldenkrais, aerobics, and physiotherapy-associated activity) and vigorous (running, cycling, swimming, ball games, and gym)	Never/past: 1.00 (reference) **Light-moderate: 0.72 (0.57–0.89)** Vigorous: 0.74 (0.54–1.0)
Willey et al. (2015) [28]	3298	11.8 years	Age > 39. Leisure time PA (LTPA) self-reported questionnaire based on prior 2 weeks	No LTPA: 1.00 (reference) **Any LTPA: 0.84 (0.75–0.94)**
Yates et al. (2008) [29]	970	16 years	Initial mean age 72, followed up until 90. Self-reported frequency of vigorous exercise sufficient to cause sweat.	Rarely/never: 1.00 (reference) **1–4 time/month: 0.78 (0.67–0.91)** **2–4 times/week: 0.72 (0.62–0.83)** **≥5 times/week: 0.81 (0.69–0.96)**
Zhou et al. (2021) [30]	11,744	5.4 years	Age ≥ 35, self-reported PA according to IPAQ	Low: 1.00 (reference) **Moderate: 0.75 (0.62–0.91)** **High: 0.48 (0.40–0.57)**

**Table 3 healthcare-13-02229-t003:** Associations with GLP-1 RA and all-cause mortality; significant findings at 95% CI ratio in bold.

Authors	Cases	Duration of Follow-Up (Average)	Exposure (No-Use Versus:)	Results (Ratios and 95% CIs)
Alenezi et al. (2024) [31]	15,655	6 months	Age > 18, GLP-1 RA prescription vs. no GLP-1RA	**0.27 (0.21–0.36)**
Huang et al. (2024) [32]	12,123	5 years	Obese individuals without T2DM on GLP-1 RA vs. no GLP-1RA	**0.23 (0.15–0.34)**

**Table 4 healthcare-13-02229-t004:** Associations between NSAIDs and all-cause mortality; significant findings at 95% CI ratio in bold.

Authors	Cases	Duration of Follow-Up (Average)	Exposure	Results (Ratios and 95% CIs)
Bardia et al. (2007) [33]	22,507	10 years	Post menopausal women age 55–69, NSAID vs. no NSAID	0.97 (0.90–1.04)
Han et al. (2020) [34]	649,339	10 years	Age > 65, ever used prescription NSAID vs. no NSAID	**0.84 (0.83–0.85)**
Liu et al. (2017) [35]	1025	8 years	Age > 50, Patient with symptomatic knee OA—NSAID use vs. no NSAID use	1.45 (0.93–21.26)
Nash et al. (2019) [36]	46,107	30 days	Age > 65, NSAID prescribed vs. no NSAID	0.83 (0.60–1.16)

**Table 6 healthcare-13-02229-t006:** All-cause mortality following total knee arthroplasty; significant findings at 95% CI ratio in bold.

Study Authors	Cases	Follow Up Mean (Range)	Control	Results (Ratios and 95% CIs)
Choi, et al. [43]	5072	58.1 months (1–12 years)	General population	**0.61 (0.54–0.70)**
Cook et al. [39]	125,367	90 days	OA population	30 days: 1.14 (0.97, 1.34) **60 days, 0.83 (0.74, 0.95)** **90 days, 0.70 (0.63, 0.78)**
Kim et al. [42]	601	13 years (10–16.5)	General population	**Over 70 years: 0.63 (0.49–0.8)** **Aged 70–79: 0.67 (0.45–0.96)** **Aged > 80: 0.69 (0.58–0.82)**
Maradit Kremers, Larson, Noureldin, Schleck, Jiranek and Berry [37]	1980	10.8 years	General population	**0.80 (0.75–0.86)**
Lizaur-Utrilla, et al. [44]	11,569	8.1 years (5–13)	General population	**86.8 (79.9–91.2)**
Lovald, Ong, Lau, Schmier, Bozic and Kurtz [16]	134,458	(1–7 years)	OA population	**1 year 0.49** **3 years 0.48** **5 years 0.50** **7 years 0.54**
Robertsson, et al. [45]	65,515	(0–28 years)	General population	**0.77 (0.76–0.78)**
Visuri et al. [46]	9443	14 years (1–33)	General population	Overall 1.00 (0.98–1.02) **0–1 year: 0.50 (0.45–0.55)** **2–9 years: 0.78 (0.75–0.81)** **10–19 years: 1.23 (1.19–1.26)** **>20 years: 1.95 (1.79–2.11)**
Yeh et al. [47]	103,114	144 months (0–17 years)	OA population	**0.791 (0.746–0.840)**
Zhou et al. [48]	22,938	11 years (0–23)	General population	**Overall: 1.08 (1.06–1.09)** **0–5 yr: 0.59 (0.57–0.60)** **>11 yr: 3.13 (2.95–3.31)**

**Table 7 healthcare-13-02229-t007:** All-cause mortality associated with opioids for non-cancer pain; significant findings at 95% CI ratio in bold.

Authors	Cases	Follow Up Mean (Range)	Exposure	Results (Ratios and 95% CIs)
Ekholm et al. [49]	13,127	11 years	Long-term and short-term opioid use for chronic pain	**Long term: 1.72 (1.23–2.41)** Short term: 1.22 (0.93–1.59) **Chronic pain and no opioid: 1.28 (1.10–1.49)**
Khodneva et al. [50]	1906	6 years	Prescription opioids	**1.15 (1.04–1.28)**
Macfarlane et al. [51]	25,864	10 years	Weak opioid and strong opioid	**Weak: 1.18 (1.06–1.33)** **Strong: 1.20 (1.01–1.43)**
Musich et al. [52]	651,556	33 months	New tramadol, new other opioid, continued tramadol, continued other opioid	**New tramadol 1.21 (1.12–1.31)** **New other opioid 1.17 (1.07–1.27). ** Continuing tramadol 1.02 (0.90–1.5) **Continuing other opioid 1.19 (1.06–1.35)**
Oh et al. [53]	10,587,018	5 years	Chronic opioid prescription, weak vs. strong opioids	**Chronic weak opioid use 0.92 (0.88–0.96)** **Chronic strong opioid use 1.35 (1.17–1.56)**
Sjøgren et al. [54]	2242	(0–8 years)	Weak opioid Strong opioid No opioid (with chronic pain)	**Chronic pain without opioids 1.21 (1.02–1.44)** Chronic pain with weak opioids 1.07 0.65–1.76) **Chronic pain with strong opioids 1.67 (1.03–2.70)**
Song et al. [55]	1,804,019	10 years	Long-term opioid	**1.21 (1.13–1.31)**

**Table 8 healthcare-13-02229-t008:** All-cause mortality associations with paracetamol; significant findings at 95% CI ratio in bold.

Authors	Cases	Follow Up Mean (Range)	Results (Ratios and 95% CIs)
Girard et al. [56]	5429	18 months	0.97 (0.86–1.10)
Lipworth et al. [57]	49,890	3.5 years (0–8)	**1.9 (1.88–1.94)**

## Data Availability

No original data was created for this study.

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
