# Peer review of "Associations Between Common Hip and Knee Osteoarthritis Treatments and All-Cause Mortality"

_healthcare, 2025, doi:10.3390/healthcare13172229_

Round 1

Reviewer 1 Report

Comments and Suggestions for Authors

Thank you for the manuscript. The topic of this review is highly relevant to the readership and involves a very common condition in clinical practice. There are however some methodological issues that must be resolved or clarified to ensure that the review is not misleading.
1. It appears that systematic review method was applied for the literature search, which is appropriate. However, systematic review requires more than one database search to make sure that the included studies represent a comprehensive body of existing work. The current search strategy uses only one database
2. The authors should justify why only observational studies are included, since this type of design carries lower weight compared to interventional trials (e.g., RCT).
3. The heterogeneity of the control populations (general population and OA population) introduces major bias to the data synthesis. This is especially true for those papers that uses non-OA population as control because this will lead to overestimation of the effects of OA treatment arms. This issue should be addressed in the discussion and, if possible, by performing subgroup analysis.

Author Response

  1. It appears that systematic review method was applied for the literature search, which is appropriate. However, systematic review requires more than one database search to make sure that the included studies represent a comprehensive body of existing work. The current search strategy uses only one database.

We accept that a systematic review should use multiple databases. However, firstly this is a structured narrative review rather than systematic review. Also, Web of Science was chosen as it is an umbrella database that includes input from multiple other databases and we have added this comment to the methods. However, we have added to the discussion of study limitations that we did not exhaust all literature with our search strategy and we accept this as a limitation.

  1. The authors should justify why only observational studies are included, since this type of design carries lower weight compared to interventional trials (e.g., RCT).

Because our endpoint was mortality, RCTs are almost never going to contain enough deaths to compare between a treatment and a control. It would be very difficult for ethics to approve a RCT expecting to find a significant difference in death rates between a treatment and a control. When trying to compare mortality rates between a treatment group and a non-treatment group, observational studies are much more likely to contain enough data to compare death rates between groups. We searched for multiple different observational study types.

  1. The heterogeneity of the control populations (general population and OA population) introduces major bias to the data synthesis. This is especially true for those papers that uses non-OA population as control because this will lead to overestimation of the effects of OA treatment arms. This issue should be addressed in the discussion and, if possible, by performing subgroup analysis.

Our study was a review paper and hence we are generally unable to perform sub-group analysis as we are not in possession of the original datasets. We have made further admission of this limitation in the discussion section. There are some subgroup analyses in Table 7 regarding Opioids

Reviewer 2 Report

Comments and Suggestions for Authors

This manuscript is a narrative review of observation studies concerning the association between the mortality and the OA treatments. The work is worth studying and should be of interest. However, some revisions and clarifications are needed as follows: 

-Abstract: I would think that stating numerical values of mortality and C.I. in the results is quite ambiguous since I am not sure how the authors came up with these values from lots of studies reviewed and different types of mortality used. Only the association or trending may be given here and the detailed numerical or statistical results should be directed to the main content in the manuscript. 

-Introduction: More details about NSAID and GLP-1 RA should be  expanded since they were the associated treatments. Currently, they were only briefly mentioned.

-Since Duloxetine is not included in the analysis due to the paucity, the content about Duloxetine in page 2 in the introduction and aims sections and section 3.8 is suggested to be removed. These sentences should also be removed. "Duloxetine was unable to be evaluated in this review due to paucity of studies. While conditionally recommended in guidelines [9,14], it lacks approval for this indication in most jurisdictions. More research may be available in the near future if its use increases. Otherwise, authors might have to provid more details and discussion about this paucity.

- The reason for the selection of Web of Sci for searching may be given. Why not MEDLINE/Pubmed, Embase, Scopus or combination of these?

-P.3, Please clarify the meaning of inception date in "A search was conducted of the Web of Science database from inception to 23 September 2024 with strategy detailed in Table 1."

-Search strategy, the keyword "SNRI" was employed, but no full name was given or described.

-P. 4, I would suggest to state which two authors screened the abstracts, which one screen the eligibility and which two reviewed the study for selection. May used the name abbreviations in the bracket.

-Fig 1, The content in the included box is incomplete. Revision is needed.

-P. 9, spelling error "hazard ration (HR)"

-P. 12, "and other treatments with no consistent associations." Such "paracetamol and NSAID" should be stated here as mentioned "other treatment" for consistency with previous sentences.

-P.12, "it is however 300 important to note the suggestion isn’t an increased risk of all-cause mortality at the highest levels" Formal word is needed here.

-P.13, "Multiple large nationwide studies, from Denmark and the United Kingdom have found an association between NSAIDs and cardiovascular mortality". Please state clearly what kind of association, increasing or decreasing.

-Some full name were used for the first time without abbreviations which were given later. Abbriviation should be given with the first time use of full name i.e Hazard Ratios (HR), Relative Risk (RR) or Standardised Mortality Ratios (SMR). Also, after givinf full name and abbreviation for the first time, there is no need to giving these again, only abbrevition should be used. Please recheck.

- Please provide some definations or explanations about the difference among HR,RR, SMR.

-Please be consistent about the statement and location of C.I.  and significant result in all tables and ensure the correctness of the statement i.e the top vs bottom location, "95% unless otherwise stated." where was the C.I. values instead of 95%?, 

- All tables, please clarify about the results presented which is the mortaility. Are they all the same for each study? what are they for each study or each table? 

-The term "opiates" was used interchangebly with "opioids" in several places. The meaning of both terms is slightly different in terms of the the origin sources. All opiates are opioids, but not all opioids are opiates. In the search strategy, author used both "opioid" and "opiate". I would suggest to clarify the terms and revise to reflect the intention and results gained from such review. I would think that the term "opioids" is more appropriate since it includes all substances (natural, semi-synthetic, or synthetic) .

-Authors stated that their review did not assess efficacy of treatments and several limitation were detailed. However, the suggestion of treatments in the Conclusion section was given. I would think that it might be softened to state only the observed outcomes without suggestion of treatments.

Author Response

-Abstract: I would think that stating numerical values of mortality and C.I. in the results is quite ambiguous since I am not sure how the authors came up with these values from lots of studies reviewed and different types of mortality used. Only the association or trending may be given here and the detailed numerical or statistical results should be directed to the main content in the manuscript

We accept that combining hazard ratios was inappropriate due to differences in methodology and that it was therefore also inappropriate to quote hazard ratios in the abstract. These have been removed and the trends noted.

-Introduction: More details about NSAID and GLP-1 RA should be expanded since they were the associated treatments. Currently, they were only briefly mentioned.

Further detail has been added.

-Since Duloxetine is not included in the analysis due to the paucity, the content about Duloxetine in page 2 in the introduction and aims sections and section 3.8 is suggested to be removed. These sentences should also be removed. "Duloxetine was unable to be evaluated in this review due to paucity of studies. While conditionally recommended in guidelines [9,14], it lacks approval for this indication in most jurisdictions. More research may be available in the near future if its use increases. Otherwise, authors might have to provide more details and discussion about this paucity.

We have removed the references to Duloxetine in the Introduction as suggested

- The reason for the selection of Web of Sci for searching may be given. Why not MEDLINE/Pubmed, Embase, Scopus or combination of these?

We chose Web of Science as an umbrella database as it contains multiple other databases within it. We have added this explanation to methods.

-P.3, Please clarify the meaning of inception date in "A search was conducted of the Web of Science database from inception to 23 September 2024 with strategy detailed in Table 1."

We simplified this to say the search was conducted on Sept 23 2024.

-Search strategy, the keyword "SNRI" was employed, but no full name was given or described.

Have added definition of SNRI to the methods.

-P. 4, I would suggest to state which two authors screened the abstracts, which one screen the eligibility and which two reviewed the study for selection. May used the name abbreviations in the bracket.

Initials have been added

-Fig 1, The content in the included box is incomplete. Revision is needed.

-P. 9, spelling error "hazard ration (HR)"

Thank you, corrected

-P. 12, "and other treatments with no consistent associations." Such "paracetamol and NSAID" should be stated here as mentioned "other treatment" for consistency with previous sentences.

Added

-P.12, "it is however 300 important to note the suggestion isn’t an increased risk of all-cause mortality at the highest levels" Formal word is needed here.

Simplified and changed sentence

-P.13, "Multiple large nationwide studies, from Denmark and the United Kingdom have found an association between NSAIDs and cardiovascular mortality". Please state clearly what kind of association, increasing or decreasing.

Increased – has been added

-Some full name were used for the first time without abbreviations which were given later. Abbreviation should be given with the first time use of full name i.e Hazard Ratios (HR), Relative Risk (RR) or Standardised Mortality Ratios (SMR). Also, after giving full name and abbreviation for the first time, there is no need to giving these again, only abbreviation should be used. Please recheck.

Have removed duplicate definitions.

- Please provide some definitions or explanations about the difference among HR,RR, SMR.

Have defined at the start of results.

-Please be consistent about the statement and location of C.I.  and significant result in all tables and ensure the correctness of the statement i.e the top vs bottom location, "95% unless otherwise stated." where was the C.I. values instead of 95%?, 

- All tables, please clarify about the results presented which is the mortaility. Are they all the same for each study? what are they for each study or each table

Have updated all tables to indicate the Ratios are all mortality and consistent for each table.

-The term "opiates" was used interchangebly with "opioids" in several places. The meaning of both terms is slightly different in terms of the the origin sources. All opiates are opioids, but not all opioids are opiates. In the search strategy, author used both "opioid" and "opiate". I would suggest to clarify the terms and revise to reflect the intention and results gained from such review. I would think that the term "opioids" is more appropriate since it includes all substances (natural, semi-synthetic, or synthetic) .

Thanks for this suggestion and we fully agree. We have changed to using the term opioids throughout.

-Authors stated that their review did not assess efficacy of treatments and several limitation were detailed. However, the suggestion of treatments in the Conclusion section was given. I would think that it might be softened to state only the observed outcomes without suggestion of treatments.

Agree, limitation added to Conclusion to state this.

Reviewer 3 Report

Comments and Suggestions for Authors

Dear Authors,

I am delighted to review this thoroughly crafted manuscript and commend the authors for delivering a clear, well‐structured synthesis of osteoarthritis treatments and their impact on long‐term mortality. The paper’s systematic approach gives readers confidence in the analysis's scope and depth. In particular, the innovative integration of emerging evidence on GLP‑1 receptor agonists offers a fresh perspective. The balanced discussion of both short‑term and long‑term mortality trajectories and thoughtful consideration of biphasic treatment effects significantly advance our understanding of how OA interventions influence patient survival.

However, I do have a few minor comments and questions that I believe will further enhance the manuscript’s clarity and readability:

Although the STROBE checklist was used to appraise included studies (pp. 2–3), no detailed quality table or scoring is presented. Please include a summary table indicating which STROBE items were met or omitted per study, so that readers can assess risk of bias component by component (e.g., confounder control, loss to follow‑up) also as material in integration.

For pharmacotherapies (NSAIDs, opioids, paracetamol), exposure is based on prescription records without information on actual dose, duration, or adherence. Acknowledge this limitation more explicitly, and—if possible—report any subgroup analyses by cumulative dose or duration performed in the primary studies.

Only two TriNetX‐derived studies (n≈13,000 each) inform the GLP‑1 RA findings, both retrospective. The term “strong” mortality reduction (HR 0.23–0.27) may overstate certainty; I suggest tempering the language and calling for prospective trials in OA.

Author Response

Although the STROBE checklist was used to appraise included studies (pp. 2–3), no detailed quality table or scoring is presented. Please include a summary table indicating which STROBE items were met or omitted per study, so that readers can assess risk of bias component by component (e.g., confounder control, loss to follow‑up) also as material in integration.

We have called our review a structured narrative review rather than a systematic review. We found that consistently all of the studies had low quality with moderate to high risk of bias and have stated this.

For pharmacotherapies (NSAIDs, opioids, paracetamol), exposure is based on prescription records without information on actual dose, duration, or adherence. Acknowledge this limitation more explicitly, and—if possible—report any subgroup analyses by cumulative dose or duration performed in the primary studies.

We have noted this as a significant limitation in what is now an expanded limitation section. In Table 7 we have provided some subgroup analysis for opioids

Only two TriNetX‐derived studies (n≈13,000 each) informed the GLP‑1 RA findings, both retrospective. The term “strong” mortality reduction (HR 0.23–0.27) may overstate certainty; I suggest tempering the language and calling for prospective trials in OA.

Agree, have deleted the term Strong from the abstract and have called for further research in the OA specific population.

Round 2

Reviewer 2 Report

Comments and Suggestions for Authors

The authors have revised the manuscript accordingly.